# Enhanced Hydrogen Evolution Reactivity of T’-Phase Tungsten Dichalcogenides (WS_2_, WSe_2_, and WTe_2_) Materials: A DFT Study

**DOI:** 10.3390/ijms231911727

**Published:** 2022-10-03

**Authors:** Haihua Huang, Guowei Hu, Chengchao Hu, Xiaofeng Fan

**Affiliations:** 1School of Materials Science and Engineering, Liaocheng University, Liaocheng 252059, China; 2College of Materials Science and Engineering, Jilin University, Changchun 130012, China

**Keywords:** first-principle calculation, electrocatalyst, hydrogen evolution reaction, electronic structure, transition metal dichalcogenides

## Abstract

The hydrogen evolution reaction (HER) plays a crucial role in hydrogen gas production. Layers of transition-metal dichalcogenides (TMDs) possess adjustable electronic structures, and TMDs with H-phase structures have been proposed as substitute HER catalysts. Nonetheless, there are few systematic theoretical analyses of the HER catalytic properties of TMDs with T’-phase structures. Using a DFT calculation, we investigated the electrocatalytic properties of W-based dichalcogenides (WS_2_, WSe_2_, and WTe_2_) through defect engineering. It was found that the interaction of H atoms with the basal plane can be tuned using non-metallic atomic doping, especially with P, thereby enhancing catalytic activity. Furthermore, the computation results demonstrated that high P-doping concentrations can enhance the number of active sites and exhibit a suitable Δ*G*_H*_.

## 1. Introduction

As an energy carrier, hydrogen (H_2_) with 142 MJ kg^−1^ maximum energy density is ideal for future energy sustainability [1,2,3]. Electrolytic water has been considered a sustainable method for converting sustainable energy into a clearer energy carrier [4]. The hydrogen evolution reaction (HER), as the cathodic half-reaction of water splitting, plays a crucial role in hydrogen gas production, which always requires a competent catalyst to lower its overpotential and accelerate reaction kinetics. Currently, platinum (Pt) and Pt-based alloys are the most excellent electrochemical catalysts [5,6]. Nonetheless, the high cost and poor availability of Pt substantially prohibit its widespread application. Hence, cost-effective, highly efficient, and earth-abundant electrocatalysts for HER are highly desirable, especially those with great promise to replace noble Pt.

Two-dimensional (2D) materials with an atomic layer thickness typically have large specific surface areas, abundant active sites, and long-term stability, which are beneficial for their catalytic properties. For 2D materials, such as transition metal-based disulfide, the original base plane is inserted for HER electrocatalysis because of the low electrical conductivity. Meanwhile, the specific surface area of the edge site, which is beneficial for the catalytic properties, is much smaller than that of the base surface. In addition, only part of the edge sites can be used in the process of HER, which greatly reduces the concentration of active sites, and thus the original 2D materials show a very low electrocatalytic performance. More researchers have found that defect engineering can not only activate the inert substrate, but also create more active sites [7,8,9].

In recent years, as a star material of 2D materials, transition metal chalcogenides (TMDs) based on molybdenum (Mo) and tungsten (W) have been progressively applied in the field of electrocatalysis due to their variable crystal structures (2H-phase, 1T-phase, and 1T’-phase) and electronic structures (semiconductor, semi-metal, and metal) [10,11,12,13]. For instance, the superior HER catalytic activity of 2H-phase TMDs has been explored and is considered a potential electrocatalyst candidate. However, the HER active sites of 2H-TMDs are located at exposed edges, which significantly restricts the concentration of active sites [14,15,16,17]. Instead, the basal planes possess large surface areas that exhibit low electronic conductivity, which limits their HER performance. For enhancing the HER activity of 2H-TMDs, it is imperative to develop effective strategies to increase the concentration of active sites and improve the electrical conductivity of the basal plane. To date, various strategies have been explored to rouse the activity of inert basal planes, such as heteroatom doping [18,19,20,21], vacancy modifications [22,23,24,25], edge-site engineering [26], and phase transition engineering [27,28,29]. For example, vacancy and defect modifications have been shown to significantly enhance the electrocatalytic performance of the basal plane in MoS_2_ [30,31]. Heteroatom (metal or non-metal element) doping has been proposed to activate the inert basal plane and increase the number of active sites [32,33]. For instance, the Fe-doped monolayer WS_2_ displayed a much lower overpotential of 195 mV than those of the pristine monolayer WS_2_ [34]. Compared with the pristine WSe_2_, the doping effect of the Co atom in WSe_2_ has been proven, and it exhibited a significant improvement in HER performance [35]. Similarly, doping WSe_2_ with various transition metals (Fe, Co, Nb, Ni, and Zr) has been undertaken, and it was found that the Ni-WSe_2_ combination is the most promising electrocatalyst for HER, which originated from its improved hydrogen adsorption [36]. Very recently, light-element (such as B, N, P, and O)-doped TMDs also exhibited superior HER activity due to their improved conductivity and enhanced H adsorption process [37,38,39,40]. In Pt- and Pd-based dichalcogenides, those that are B-doped have been reported to have optimized electronic structures, raising the activity level and providing sufficient active sites on the basal plane [32]. The N-doped WSe_2_ nanosheets prepared by Sun et al. exhibited a low onset potential of 86 mV in the HER process, mainly due to the introduction of localized charges at the Fermi level through the N-doping, enhancing the interaction with H [41]. Further, the development of new 2D materials has become a feasible method to find effective catalysts [42,43,44,45].

Alternatively, T’-phase TMDs with metallic characteristics have considerably high catalytic activity resulting from their high electron conductivity and abundant active sites both at the surface areas and edges [46,47,48,49]. Thus, phase transition engineering has become an effective pathway to improve catalytic performance [27,28,29,50]. Theoretical analysis results have displayed that the T’-TMDs are a potential candidate catalyst material for HER, especially for T’-MoSe_2_ and T’-WS_2_ with a hydrogen adsorption free energy (Δ*G*_H*_) of −0.04 eV and 0.03 eV, respectively [51,52]. Experimental studies and theoretical simulation conclusions have demonstrated that the basal plane of metallic T’-TMDs indeed shows high catalytic activity, which is similar to that of a Platinum catalyst [53,54,55,56]. To further improve the catalytic activity of the T’ phase structure, constructing defect structures (such as chalcogen vacancies) [57,58] and plasma treatments [59] have been confirmed to be effective methods to optimize electrochemical catalytic activity. Although there is measurable progress in both the experimental and theoretical research on T’-TMD electrocatalysts for HER, the systematic study of the catalytic mechanism of T’-TMDs toward the HER is largely unexplored. Hence, it is vital to analyze the active sites and the hydrogen evolution mechanism of the T’ phase in detail.

Compared with Mo, W is more abundant in the Earth’s crust and its derivates are more environmentally friendly. Meanwhile, the direct synthesis of T’-WSe_2_ and WTe_2_ has been experimentally achieved [60]. Thus, we turn our attention to the W-based dichalcogenides (WX_2_, X = S, Se, and Te) with a T’-phase, where we expect that the defect engineering will provide abundant active sites and achieve further improved HER catalytic performance. In our work, we first calculate the feasibility of different defect configurations in the WX_2_ monolayer, including defect (single-vacancy and double-vacancy) and light element (B, C, N, and P) doping. Then, the influence of the defects on the electronic structure is analyzed. Through the analysis of the electronic structure, the mechanism of interaction between the basal plane with defects and H is investigated. Finally, we estimate the Gibbs free energy of H adsorption (Δ*G*_H*_) and assess how different defect structures affect the HER catalytic activity. The results of our calculations suggest that the light-element-doping, especially with P, is a feasible method for improving the HER catalytic performance of T’-TMDs.

## 2. Computational Details

All calculations in our work are based on the density functional theory (DFT) and realized through the Vienna ab initio simulation package (VASP) [61]. The exchange–correlation interactions between electrons are described by the generalized gradient approximation (GGA) of the parameterization of Perdew, Burke, and Ernzerh (PBE) [62]. We chose the plane-wave cutoff energy of 550 eV to ensure the convergence of the total energy. The geometries were optimized until the Hellmann–Feynman residual force per atom converged to less than 0.01 eV/Å and achieved the convergence criteria of 10^−4^ eV per atom. To ignore the interaction of the neighboring layers, we chose the vacuum layer that was larger than 18 Å.

The primitive cell structure of the T’-TMDs is rectangular and the structural parameters we calculated are shown in Table 1, which correspond to the previous calculation results [63]. To illustrate the defects’ effects on the electrocatalytic properties of the T’-WX_2_ (X = S, Se, and Te), we constructed six different kinds of defect configurations, including the single-vacancy of the chalcogen atom at site 1 (V1) and site 2 (V2), the double-vacancy of the chalcogen atom at site 1 (DV1) and site 2 (DV2), and light element (B, C, N, and P) doping at site 1 and site 2. We expected that the vacancy and doping defect would introduce more electron states at the Fermi surface to enhance the electrical activity. Schematic diagrams of the structure models with different defect configurations are displayed in Figure 1. All the defect structures we considered are constructed in 4 × 2 × 1 supercells.

Firstly, we calculated the formation energy of the defects to evaluate the thermodynamic stability of the various defect configurations. The formation energy of the defect structures is computed through the following equation:(1)$$E_{f}=E_{D}-E_{P}+\Delta n_{i}\sum_{i}\mu_{i}$$
where *E_D_* and *E_P_* denote the total energy of the MX_2_ monolayer with the defect structure and the pristine structure, respectively, Δ*n_i_* is the difference in several atoms of type *i* between those without and with the defect structure, and *µ_i_* represents the energy of atom type *i*, which corresponds to the energy in its bulk counterpart.

The H adsorption energy (*E*_ads_) on the different active sites of the basal surface allows us to analyze the interaction of H and the catalyst to obtain insight into the HER catalytic characteristics, and it is calculated as follows:(2)$$E_{\mathrm{ads}}=E_{H+\mathrm{cat}}-E_{\mathrm{cat}}-E_{H}$$
where *E*_H+cat_ and *E*_cat_ represent the total energy of the MX_2_ hydrogen adsorption system and catalyst, respectively, and *E*_H_ is the energy of the isolated H atom.

Norskov et al. [6] proposed that the free energy of the (H^+^ + e^−^) system is replaced by half of the chemical potential of a hydrogen molecule. Therefore, according to the HER process (${2H}^{*}+2e^{-}\to H_{2}$), the free energy of the intermediate state H* is critical for the reaction barrier. As a commonly significant indicator of HER activity, the Gibbs free energy Δ*G*_H*_ can be expressed using the following formula:(3)$$\Delta G_{H*}=E_{H+\mathrm{cat}}-E_{\mathrm{cat}}-1/2E_{H_{2}}+\Delta E_{\mathrm{ZPE}}-T\Delta S$$
where $E_{H_{2}}$ is the energy of the free H_2_ molecule and Δ*E*_ZPE_ and Δ*S* represent the difference in the zero-point energy and the entropy between the final state and adsorption state H*, respectively. The contribution of vibrational entropy from H in the adsorption state H* can be ignored. Thus, we can conclude that Δ*S* can be obtained based on the entropy of the gas phase at standard conditions. At 300 K, the *T*Δ*S* K is approximately −0.2 eV. In addition, the contribution of Δ*E*_ZPE_ to the free energy for H adsorption is typically small. A reasonable conclusion can be obtained by using the Δ*E*_ZPE_ of the H adsorbed on the Cu (111) surface (0.04 eV) as an approximation of the adsorption system. Thus, the Δ*E*_ZPE_ plus *T*Δ*S* can be chosen for 0.24 eV in the HER process.

To an extent, the theoretical exchange current density (*i*_0_) can reflect the rate of the proton transfer to the surface, which can be obtained on the basis of Δ*G*_H*_. The detailed calculation equation at standard conditions (pH = 0 and *T* = 300 K) is as follows.

When the proton transfer is exothermic (Δ*G*_H*_ < 0), the *i*_0_ can be described as [6]
(4)$$i_{0}=\frac{-ek_{0}}{1+\exp\left(\frac{-\Delta G_{H*}}{kT}\right)}.$$

If the reaction process is endothermic (Δ*G*_H*_ > 0), the *i*_0_ is computed by
(5)$$i_{0}=\frac{-ek_{0}\exp\left(\frac{-\Delta G_{H*}}{kT}\right)}{1+\exp\left(\frac{-\Delta G_{H*}}{kT}\right)},$$
where *k*_0_ is the rate constant (200 s^−1^/site) and *k* denotes the Boltzmann constant.

## 3. Results and Discussion

### 3.1. Defect Structures and H Adsorption

As a typical defect type, the vacancy is very common in two-dimensional materials. We first calculated the defect formation energy of a non-metal single-vacancy (V) and a double-vacancy (DV), as shown in Figure 2a. In terms of the formation energy calculated with Equation (1), the V and DV easily form due to the negative formation energy for all monolayer T’-WX_2_s (X = S, Se, and Te), which suggested that they are feasible defect types during the experiment synthesis. In addition, non-metal doping is also an effective method to tune the electronic structure and improve the electrocatalytic properties of the materials. Since the chalcogen atom is located in the out layer of the WX_2_, it is easily replaced by dopant atoms. Therefore, we calculated and obtained the most stable doping position for the different non-metal doping atoms (B, C, N, and P) in the doped systems. The calculated defect formation energies at the most stable sites are shown in Figure 2a. Apparently, the formation energy of N-doping is positive, which means that the N substitution of chalcogen atoms is not experimentally feasible. By contrast, the B, C, and P substitutions become feasible because of their low defect formation energy.

Then, we investigated the hydrogen (H) adsorption process on the surface of the six selected configurations, including a pristine structure, the single-vacancy (V), the double-vacancy (DV), and the B-, C-, and P-doping of the WS_2_, WSe_2_, and WTe_2_ monolayers, respectively. Hydrogen tends to be adsorbed on a site with a lower adsorption energy, and so various H adsorption sites were considered for each structure. The results are listed in Appendix A. Similar to other 2D materials, H preferred to absorb on the top of the chalcogen atom for all the materials we considered. For the single-vacancy model, the vacancy tended to appear at site 2 (marked in Figure 1). Compared with the H adsorption energy, we concluded that the adsorbed H atom preferred to occupy the top side of the X (S, Se, and Te) vacancy for the WX_2_. In the case of the double-vacancy, the stable double-vacancy configuration was DV2 (marked in Figure 1) and the low-energy adsorbed H atom was located at the middle site of the double vacancy. For the non-metal doping model, the dopant tended to occupy the chalcogen atom at site 2. By calculating the H adsorption energies at different locations, including the top site of the dopant and the adjacent chalcogen atoms, we found that the H tended to adsorb on the top side of the dopant.

The H adsorption energies (*E*_ads_) at the most stable positions are presented in Figure 2b. The results of the adsorption energies for the various defects suggested that the interaction between the defects and H is stronger than that of the pristine WX_2_ and H. For the pristine T’-WX_2_, the adsorption ability decreased sequentially for S and Se to Te. This is mainly due to the difference in element electronegativity. Among the considered various defect models, the adsorption energy of C-doping is the largest, which means the interaction between C and H is strong in C-doped WS_2_, WSe_2_, and WTe_2_. The interaction strength between other defect types (single-vacancy, double-vacancy, and B-doping) and H was similar. For the P-doped T’-WX_2_, the interaction of H and P was stronger than that of the pristine base plane with H, but it was weaker than that of the other defect configurations (including the single-vacancy, double-vacancy, and non-metal (B and C) doping).

### 3.2. Electronic Structures of the Defective Configurations

To elucidate the underlying mechanism of the interaction between H and the defects, we calculated the electronic structures of the various defect configurations. The electronic density of the states of the pristine T’-WX_2_ (X = S, Se, and Te) in Appendix A suggested that the WX_2_ with a T’-phase has metallic properties. Taking T’-WTe_2_ as an example, we analyzed the electronic structure changes after introducing different types of defects. The partial density of states (PDOS) of the single-vacancy (V), double-vacancy (DV), and B-, C-, and P-doping for the WTe_2_ are plotted in Appendix A. Compared with the pristine electronic structure, the electronic structure near the Fermi level was redistributed after the introduction of the defects. The band gap appeared after introducing the single- and double-vacancies, which means the electronic conductivity became worse (shown in Appendix A). For non-metal (B, C, and P) doping, the weak electronic states at the Fermi level were from the contribution of the coupling between the W_d orbital and the B/C/P_p orbital. This can also be observed from the charge density in real space, as presented in Figure 3. The differences in electronegativity led to the different charge accumulations between the W and the dopant.

With H adsorption, the electronic state is modulated due to the presence of the charge transfer. For B/C/P-doped WX_2_ (X = S, Se, and Te), the optimal active site for the adsorption of atomic H is the dopant site as it exhibits an intensive occurrence of *sp*-hybridization. Figure 4 demonstrates the PDOS after H adsorption for the B/C/P-doped WX_2_ (X = S, Se, and Te). The *sp* interaction of the H_*s* orbital and the B/C/P-*p* orbital forms a bonding state *σ*, which is positioned lower than the Fermi level, and an anti-bonding state *σ**, which is higher than the Fermi level. Jiang et al. [64] reported that the energy level of *sp*-hybridization with the most intensity (*sp*_max_) below the Fermi level is the key descriptor to denote the interaction of H and the catalyst. We extracted the *sp*_max_ values of various adsorption systems. Herein, the *sp*_max_ values are −6.78, −6.57, and −6.28 eV for the B-doped WS_2_, WSe_2_, and WTe_2_, respectively. For the C-doped system, the *sp*_max_ values are −6.06, −6.08, and −6.13 eV for WS_2_, WSe_2_, and WTe_2_, respectively. In the P-doped adsorption system, the *sp*_max_ values are −6.63, −6.61, and −6.56 eV, respectively. Apparently, the P-doped T’-WX_2_ has a deeper *sp*-hybridization energy level compared to the doping systems we considered, meaning that more electrons were filled into anti-bonding and caused the weaker interaction between adsorption H and the P-doped catalyst. For the B- and C-doped WX_2_, the *sp*_max_ values are large, suggesting a stronger H adsorption, especially for C-doping. In addition, we plotted the difference in the charge density of the WTe_2_ with the various defect configurations after H adsorption. The difference in charge density is defined as follows:(6)$$\Delta \rho =\rho_{\mathrm{sub}+H}-\rho_{\mathrm{sub}}-\rho_{H}$$
where *ρ*_sub+H_, *ρ*_sub_, and *ρ*_H_ represent the charge density of a catalyst with H adsorption, a catalyst without H adsorption, and an H atom, respectively. From the difference in the charge density of the T’-WX_2_ adsorption system in Figure 5, the H atom gained electrons from the catalyst, regardless of the defect type. Therefore, it was unlike the mechanism of H-adsorption on the basal plane of H-MoS_2_ in which the electron charge is transferred from the absorbed hydrogen to the base plane. A similar phenomenon occurred in WS_2_ and WSe_2_, as shown in Appendix A.

### 3.3. HER Activity of the Defective Structures

According to Sabatier’s principle, the Gibbs free energy (Δ*G*_H*_) of the H adsorption state is a key factor to evaluate the HER catalytic activity of a catalyst. The more positive (negative) the Δ*G*_H*_, the weaker (stronger) the interaction between the H and the catalyst. As the value of Δ*G*_H*_ approaches zero, the catalysts show optimal catalytic activity. The Δ*G*_H*_ value we calculated for the WX_2_ with the different defect configurations are depicted in Figure 6a. For pristine WS_2_, WSe_2_, and WTe_2_ monolayers, the Δ*G*_H*_ values are 0.26 eV, 0.75 eV, and 0.95 eV, respectively. These positive Δ*G*_H*_ values indicate a weaker adsorption of H on the catalyst surface and make the Volmer adsorption process infeasible. Compared with the pristine basal plane, non-metal atom doping can significantly decrease the Δ*G*_H*_ and improve the adsorption strength. Among the B/C/P doping systems, the Δ*G*_H*_ of the C-doped WX_2_ (X = S, Se, and Te) is the most negative, suggesting that the strongest interaction is between the H and the catalyst and that it is unfeasible to release H_2_ effectively. This is consistent with the analysis results from the perspective of an electronic structure. The best among those we considered for the various defects is the P-doping system, which exhibited a Δ*G*_H*_ of −0.37 eV, −0.26 eV, and −0.17 eV for the WS_2_, WSe_2_, and WTe_2_ monolayers, respectively. While the Δ*G*_H*_ value of the P-doped WTe_2_ was the closest to being thermoneutral, it exhibited promising HER catalytic activity.

As described above, the theoretical exchange current density (*i*_0_) can directly reflect the catalytic activity of a catalyst. We calculated the *i*_0_ with the relationship of the Δ*G*_H*_ (volcano curve) to compare the HER catalytic properties of the various defect configurations in the T’-WX_2_ (X = S, Se, and Te) monolayers, as depicted in Figure 6b. We concluded that the current density of the P-doped WX_2_ is higher and close to commercial Pt, which originated from the smaller value of the Δ*G*_H*_. In addition, the pristine T’-WS_2_ showed a relatively higher current density compared to the other configurations with defects. Therefore, the theoretical analysis results revealed that P-doping is an effective strategy to improve the catalytic activity of the basal plane for T’-phase W-based dichalcogenides.

Moreover, we analyzed the effects of the defect concentrations of P-doping on catalytic properties. The Δ*G*_H*_ of the different P-doping concentrations for the WS_2_, WSe_2_, and WTe_2_ monolayers are plotted in Figure 7. With the increase in P-doping concentration, the value of Δ*G*_H*_ decreases slightly. These results indicate that the P-doped WX_2_ has excellent catalytic activity, regardless of the doping concentration.

## 4. Conclusions

The modulation of the HER catalytic activity for the T’-WX_2_ monolayer has been investigated systematically by using density functional theory calculations. As expected, the T’-WX_2_ (X = S, Se, and Te) have metallic properties. According to the calculations of the hydrogen adsorption Gibbs free energy (Δ*G*_H*_), we found that the Δ*G*_H*_ of the WSe_2_ and WTe_2_ monolayers was higher than that of WS_2_, which means that WSe_2_ and WTe_2_ have a weaker basal catalytic activity. To enhance the basal activity, we modulated the electronic properties through single-vacancy, double-vacancy, and non-metallic atom doping (primarily including B, C, N, and P). We demonstrated that catalytic activity could be improved by P-doping in all the strategies we considered. The improvement in electrocatalytic activity was revealed to come from the proper hydrogen affinity of the P atom. Therefore, these results indicate that metallic T’-phase WX_2_ (X = S, Se, and Te) acts as a potential candidate for electrocatalytic application, and they provide ideas for enhancing the catalytic activity of other T’-TMDs with metallic properties.

## Figures and Tables

**Figure 1 ijms-23-11727-f001:**
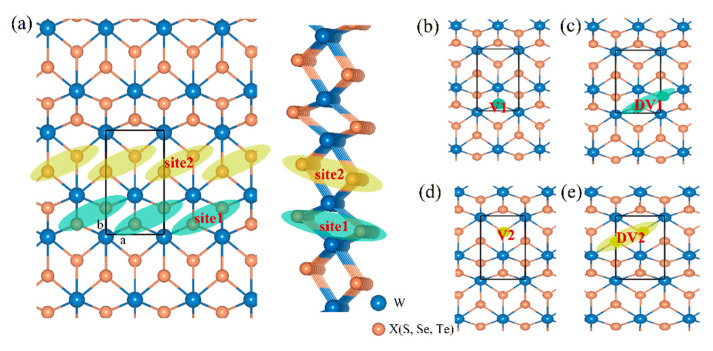
Crystal structure of the T’-WX_2_ (X = S, Se, and Te) monolayer with (**a**) a pristine structure, (**b**) the single vacancy of X at site 1 (V1), (**c**) the double vacancy of X at site 1 (DV1), (**d**) the single vacancy of X at site 2 (V2), and (**e**) the double vacancy of X at site 2 (DV2).

**Figure 2 ijms-23-11727-f002:**
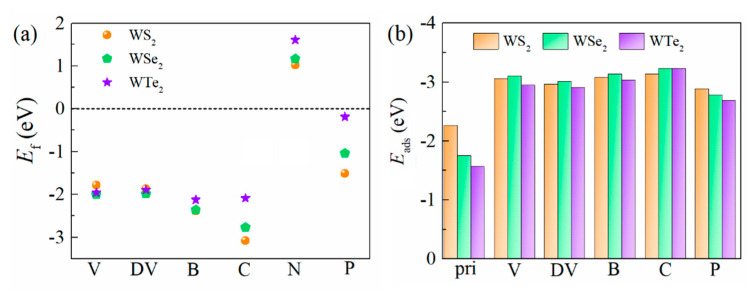
(**a**) Defect formation energy of the single−vacancy (V), double−vacancy (DV), and light element substitution for the different materials, including the T’-WS_2_, T’-WSe_2_, and T’-WTe_2_ monolayers. (**b**) Adsorption energy of H with the different defect structures in the most stable configurations.

**Figure 3 ijms-23-11727-f003:**
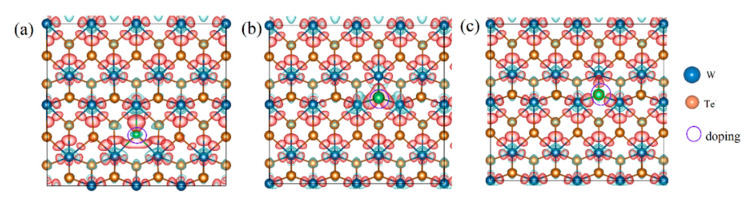
Differential charge density of the WTe_2_ with (**a**) B-, (**b**) C-, and (**c**) P-doping, respectively. The red color indicates charge accumulation and green denotes charge depletion, while the purple circles denote doping sites.

**Figure 4 ijms-23-11727-f004:**
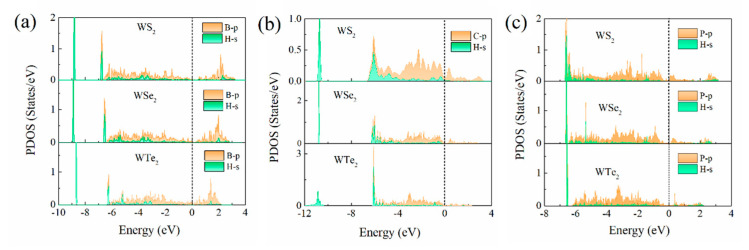
The partial density of states (PDOS) between the H_s orbital and the B/C/P_p orbital for (**a**) B-, (**b**) C-, and (**c**) P-doped T’-WS_2_, WSe_2_, and WTe_2_, respectively. The vertical dotted lines denote the Fermi level.

**Figure 5 ijms-23-11727-f005:**
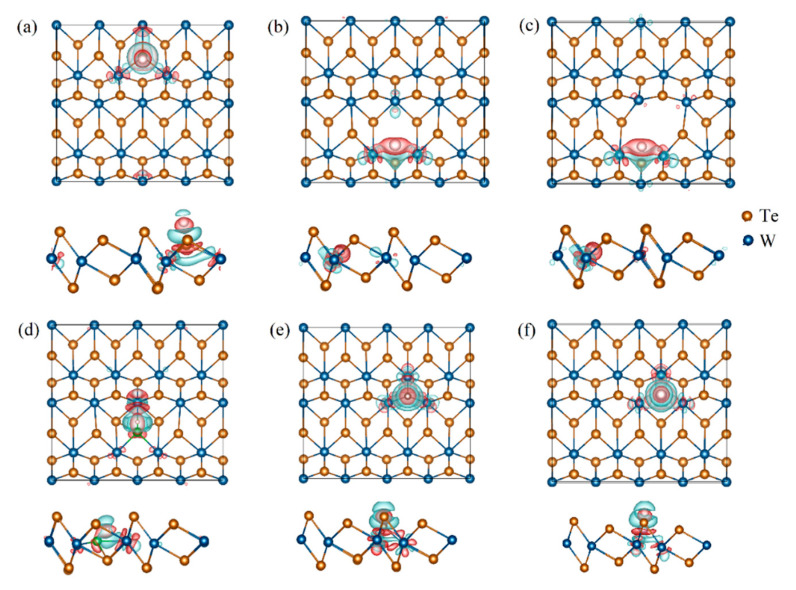
Difference in the charge densities of WTe_2_ with an H adsorption system for (**a**) a pristine, (**b**) the single-vacancy, (**c**) the double-vacancy, and the (**d**) B-, (**e**) C-, and (**f**) P-doping from the top and side view, respectively. The red color indicates charge accumulation and green denotes charge depletion.

**Figure 6 ijms-23-11727-f006:**
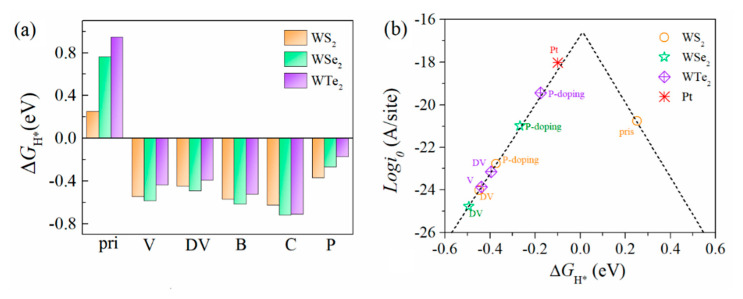
(**a**) Values of Δ*G*_H*_ and (**b**) the volcano curve between the exchange current density *i*_0_ and the Δ*G*_H*_ for the T’-WS_2_, WSe_2_, and WTe_2_ monolayers with the various defect types.

**Figure 7 ijms-23-11727-f007:**
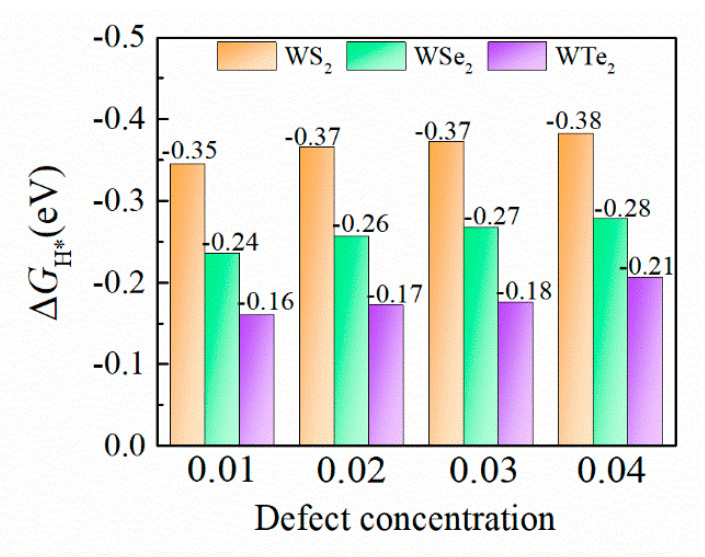
Δ*G*_H*_ values of various the P-doping concentrations for the T’-WS_2_, WSe_2_, and WTe_2_ monolayers.

**Table 1 ijms-23-11727-t001:** Lattice parameters a and b of the 1T’ W-based dichalcogenides.

Materials	Lattice	
	a (Å)	b (Å)
WS_2_	3.19	5.72
WSe_2_	3.30	5.94
WTe_2_	3.49	6.31

## Data Availability

The raw/processed data generated in this work are available upon request from the corresponding author.

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
