# Peer review of "Enhanced Hydrogen Evolution Reactivity of T’-Phase Tungsten Dichalcogenides (WS2, WSe2, and WTe2) Materials: A DFT Study"

_ijms, 2022, doi:10.3390/ijms231911727_

Round 1
Reviewer 1 Report
In this work, authors systematically investigated the electrocatalytic properties of W-based dichalcogenides (WS2, WSe2and WTe2) with T’-phase through defect engineering, and reported that the high p-type doping concentration could enhance the number of active sites. This is an interesting paper on W-based dichalcogenides in regards to its potential as electrocatalyst. It is well presented and the results are interesting. I recommend it can be published in the International Journal of Molecular Sciences after minor revision.
1. Some description in the Computational Method part is ambiguous. For example, the force convergence criterion is 10-2eV/Å. As far as I know, this value can be chosen negative or positive. If this value is positive, 10-2 is much coarser for a good convergence. The author should elucidate the sign of the value.
2. As we all known, the TMDs with T’-phase structure possess metallic character, which is beneficial to enhance the electrocatalytic activity. It is reported that the catalytic properties are improved theoretically after introducing the defect. Thus, I wonder if the author can find experimental data to prove that light element doping is an effective method to improve the electrocatalytic performance.
3. Some typos are found in the manuscript, please check, and remedy them.
Author Response
Q1. Some description in the Computational Method part is ambiguous. For example, the force convergence criterion is 10-2eV/Å. As far as I know, this value can be chosen negative or positive. If this value is positive, 10-2 is much coarser for a good convergence. The author should elucidate the sign of the value.
Response&Changes:
Thanks for the suggestion. If the force convergence criteria are positive, it refers to the sum of the forces on each atom. If it is negative, that is the force on each atom. We chose the force criteria for each atom in our calculation. This point has been elucidated in the revised manuscript.
Q2. As we all known, the TMDs with T’-phase structure possess metallic character, which is beneficial to enhance the electrocatalytic activity. It is reported that the catalytic properties are improved theoretically after introducing the defect. Thus, I wonder if the author can find experimental data to prove that light element doping is an effective method to improve the electrocatalytic performance.
Response&Changes:
In the manuscript, we have cited relevant experimental work on the improvement of catalytic properties by light element doping, such as N-doped WS2 (Nanoscale, 2020, 12, 22541-22550), O-doped WS2 (ACS Catal.,2020, 10, 6753-6762.), P-doped WS2 (Chem. Eng. J.,2022, 429, 132187; Nano Res.,2022, 15, 2855-2861).
Q3. Some typos are found in the manuscript, please check, and remedy them.
Response&Changes:
This point has been corrected in the revised manuscript.

Reviewer 2 Report
In this paper, the authors present a very interesting study about Hydrogen evolution reaction using tungsten dichacogenides. The paper is well-structured, well-written, and the Figures are clear. Given the importance of the topic, I would recommend the publication of the paper in International Journal of Molecular Sciences. I strongly recommend discussing the following points:
Q1: The paper will gain impact if the authors comment the most common drawbacks of 2D materials from the experimental point of view.
Q2: The paper would gain impact if the authors compare in the introduction these 2D materials with other 2D materials containing Mo like MXenes and the 3D version, carbides towards H adsorption (10.1038/s41699-021-00259-4, 10.1016/j.susc.2016.10.001, 10.1021/acscatal.0c03106 10.1088/1361-6463/ac8b1a among others)
Author Response
Q1: The paper will gain impact if the authors comment the most common drawbacks of 2D materials from the experimental point of view.
Response&Changes:
Thank you very much for your suggestion. We have added the related on the drawbacks of 2D materials and related references in the revised manuscript. The details as follows: “For 2D materials, such as transition metal-based disulfide, the original base plane is insert for HER electrocatalysis because of the low electrical conductivity. Meanwhile, the specific surface area of the edge site, which is beneficial for the catalytic, is much smaller than that of the base surface. Besides, only part of the edge sites can be used in the process of HER, which greatly reduces the concentration of active sites, and thus the original 2D materials show very low electrocatalytic performance. More researchers have found that defect engineering can not only activate the inert substrate, but also create more active sites (Chinese Journal of Catalysis, 43, 2022, 636–678; J. Mater. Chem. A, 2015, 3, 15927–15934. Nanoscale, 2017, 9, 16616–16621).”
Q2: The paper would gain impact if the authors compare in the introduction these 2D materials with other 2D materials containing Mo like MXenes and the 3D version, carbides towards H adsorption (10.1038/s41699-021-00259-4, 10.1016/j.susc.2016.10.001, 10.1021/acscatal.0c03106 10.1088/1361-6463/ac8b1a among others)
Response&Changes:
Thanks for the suggestion. We have compared Mo-based 2D materials (such as MXenes) with other materials and added the related references in the revised manuscript.
